# Urate Biology and Biochemistry: A Year in Review 2022

**Rachel D. King** [1] and **Eric E. Kelley** [2],*

[1] Department of Biochemistry, School of Medicine, West Virginia University, Morgantown, WV 15902, USA
[2] Department of Physiology and Pharmacology, School of Medicine, West Virginia University, Morgantown, WV 15902, USA
* Correspondence: eric.kelley@hsc.wvu.edu

**Abstract:** The past year generated significant change and advancement of the urate field with novel insights regarding the role of uric acid (UA) in multiple pathophysiologic processes from gout to COVID-19. While these contributions continue to move the field forward, the basic biochemistry and biology of UA is often overlooked, being lost in the shadow of clinical associations and omics. However, the seminal impact of UA begins with biochemistry and the associated interplay with cell biology. In these basic reactions and resultant impacts on physiology, UA mediates its influence on clinical outcomes. As such, this review focuses on published advances in UA biochemistry and biology in 2022 and associates these advances with downstream consequences.

**Keywords:** uric acid; uricase; xanthine; hypoxanthine; xanthine oxidoreductase

## 1. Introduction

Uric acid (UA) is the final product of purine catabolism in humans and greater apes. It is derived from the oxidation of xanthine by the molybdopterin-flavin enzyme, xanthine oxidoreductase (XOR) (Figure 1). Under healthy, normal physiologic conditions, the site of greatest XOR specific activity and thus UA formation is the liver and thus this organ has been seen as the primary source of UA and potentially the master supplier of UA to the circulation. However, a recent report has demonstrated that other tissues are also contributory in maintaining circulating levels of UA, as a murine model of liver-specific XOR ablation demonstrates only a 50% diminution of plasma UA concentration [1]. While this may be the case, it is important to note that several factors coalesce to determine the concentration of UA in the circulation. For example, nonprimate mammals catabolize purines one step beyond UA to allantoin by expressing urate oxidase (UA → allantoin + $H_2O_2$). In addition, UA concentration in the blood is regulated by the action of urate transporters primarily in the kidney and intestines. As such, a realistic assessment of factors that contribute to alterations in UA levels in the circulation must consider contributions from all these players.

The normal levels of uric acid in the blood range from 1.5–6 mg/dL in women, and 2.5–7 mg/dL in males, or 119–357 μM [2]. Once this range is exceeded, hyperuricemia is realized and the risk for gout is significantly elevated. Gout occurs when UA levels exceed saturation resulting in monosodium urate crystal formation, specifically in the synovial fluid bathing the joints. These crystals, when present in the joints, induce considerable inflammation and severe pain. Dogmatically, it has been assumed for years that conditions which favor gout are generally dietary where consumption of purine-rich food results in absorption of excess purine that must then be catabolized. This excess in substrates for the purine catabolic pathways ultimately leads to enhanced activity of XOR to generate UA. However, as stated above, alterations in clearance capacity (urate transporters) in the kidneys can also contribute to this process. Interestingly, a recent report has noted a critical association between circulating ferritin levels and UA concentration and thus the potential for iron to mediate elevation in XOR and by extension UA levels [3]. While

mechanistic details driving this process are not yet known, iron has been previously shown to upregulate XOR expression and enzymatic activity in both cells and tissues [4].

# Xanthine Oxidoreductase (XOR)

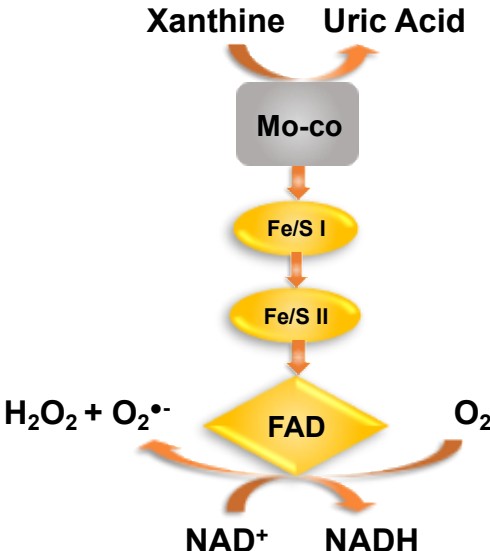

**Figure 1.** Xanthine oxidoreductase (XOR). A cartoon representation of 1 monomer of the homodimer XOR. Oxidation of hypoxanthine to xanthine and xanthine to UA occurs at the molybdenum cofactor (Mo-co). Electrons derived from this process are passed to the FAD via 2 nonidentical Fe/S centers. As xanthine dehydrogenase, the enzyme reduces $NAD^+$ to NADH and in the xanthine oxidase configuration, oxygen is reduced to peroxide and superoxide. As both the oxidase (XO) and dehydrogenase (XDH) utilize xanthine as an electron source and thus produce UA, we have simplified our description to include both isoforms of the enzyme and thus refer to it as XOR.

Hyperuricemia is also closely associated with obesity/metabolic syndrome and diabetes. As such, there is considerable focus on elevated UA being causative in the metabolic abnormalities associated with the obese/diabetic phenotype. This concept has been studied most extensively in murine models of genetically- or diet-induced obesity often coupled to fructose supplementation. Reports have demonstrated considerable elevation in UA levels as a consequence of fructose catabolism and subsequent pro-inflammatory signaling that drives insulin resistance [5–8]. Although considerable evidence has been accumulated to support causation, other reports have demonstrated that maintenance of normal UA levels in male mice does not alter the metabolic consequences of diet-induced obesity (60% HFD) (hepatocyte *XDH* KO mice on a C57Bl/6J background) and diet-induced obese (60% HFD) female mice (C57Bl/6J) with attendant metabolic dysfunction in the absence of hyperuricemia [1,9]. A critical obstacle to assigning causation versus correlation for UA and diobesity is the considerable variability in models, diets, and measured outcomes. For example, the models used thus far to interrogate this issue have consisted of genetic (e.g., pound mouse) obesity and diet-induced obesity using various diets from the 42% to 60% calories derived from fat to high-fructose or fructose in the drinking water [1,5,6,9–11]. The multiplicity of variables across many studies injects substantive difficulty in establishing trends and consensus. Therefore, work continues toward resolving this issue.

For decades, the biologic relevance of the evolutionary loss of urate oxidase expression concomitant with the loss of capacity to synthesize ascorbate (vitamin C) has been thought to assign a crucial role for UA as an antioxidant. Indeed, UA demonstrates the capacity to quench singlet oxygen, inhibit lipid peroxidation, and diminish the redox po-

tential of peroxynitrite (O=NOO$^-$) [12,13]. In fact, it has been proposed that UA may be a pseudo-replacement for endogenously supplied ascorbate and thus a primary circulating antioxidant [12]. Yet, it is important to note that UA has not been shown to react to any appreciable extent with $H_2O_2$ or superoxide (O$_2^{\bullet-}$) [12]. Uric acid can indeed act as a direct antioxidant; however, a commonly overlooked characteristic of UA is that it demonstrates significant potential for participating in indirect antioxidant actions. Whereas UA is a product of XOR turnover with xanthine, it can participate in product-based inhibition of XOR [14,15]. This property of UA has been demonstrated in both buffer solution and human plasma with an $EC_{50}$ = (~300 μM). This $EC_{50}$ value is important as it represents the upper end of the normal human-circulating plasma UA concentrations and thus XOR inhibition would be expected to be operative under physiologic conditions. This indirect antioxidant potential for UA would be considerable in the context of inflammatory processes where the major isoform of XOR would be xanthine oxidase (XO), the oxidant ($H_2O_2$ and O$_2^{\bullet-}$)-generating configuration. As for every biologic process, location and concentration will drive effects and this is no less the case for UA. As such, the antioxidant versus pro-inflammatory role for UA will depend heavily on its location (inter/extracellular) and concentration.

In the following paragraphs we will review the novel biochemical and basic biologic findings over the past year. In doing so we will reconnect with the concepts above to provide insight to where, when, and why UA plays salutary versus deleterious roles.

## 2. UA and Gout

Novel findings regarding the basic biochemistry/biology of UA in the context of gout published over the past year were not abundant but do include a report demonstrating a role for glucagon-like peptide-1 (GLP-1) and its receptor GLP-1R in the inflammatory response to monosodium urate crystals [16]. This study demonstrated GLP-1R expression in murine macrophages with greater expression in M2 and Ly6C+ macrophages than their corresponding counterparts. Ablation of GLP-1R resulted in a diminished capacity for migration concomitant with elevated expression of IL-6. When subjected to monosodium urate crystal-mediated peritonitis, mice with knockout of GLP-1R demonstrated significantly less macrophage infiltration, suggesting GLP-1R may play a pivotal role in the immune response to urate crystal-induced inflammation.

While it has been accepted that alterations in the gut microbiome can impact hyperuricemia and gout, there is a considerable gap in understanding the mechanistic linkages. In the past year, a study was published that attempted to narrow this gap using the murine model whereby urate oxidase was genetically ablated (Uox-KO) [17]. As expected, these mice are hyperuricemic and suffer from urate nephropathy. When compared to wild-type controls, the Uox-KO mice displayed a significantly altered gut microbiome, such as depletion in branched chain amino acids and dysregulation of amino acid metabolism, resulting in impaired intestinal integrity shifts in solute carrier profiles that coalesced to exacerbate the hyperuricemia. They reported an expansion (relative abundance) of the *Firmicutes* phylum and a diminution (relative abundance) of the *Verrucomicrobiota* phylum in Uox-KO mice versus controls. In addition, cohousing WT mice with Uox KO mice resulted in the WT mice assuming increased plasma UA levels compared to WT housed with other WT mice. The strains that were associated with the elevated UA levels were *Ileibacterium*, *Anaerotruncus,* and *Roseburia*, and were elevated in cohoused WT mice but were unaltered in cohoused KO mice. Results from this study reveal the potential for modulating the gut microbiome as an adjuvant therapeutic strategy for hyperuricemia.

## 3. UA and Metabolic Dysfunction

As outlined briefly above, there is a close association between circulating UA levels and obesity in both rodent models and the clinic. Much attention has recently been focused on the impact of fructose in the metabolic consequences of obesity including insulin resistance, nonalcoholic fatty liver disease (NAFLD)/nonalcoholic hepatic steatosis

(NASH), hypertension, and altered reactivity of the immune system [18]. This past year has seen advances in the field that describe the interplay between the gut microbiome and response to acute fructose challenge. A small clinical study of 57 young and middle-aged adults endeavored to evaluate the impact of acute fructose administration on serum UA levels from 30 to 180 min in the context of 30 different urate-related SNPs [19]. Three of these thirty reached significance statistically and include *HNF4G*, *INHBB*, and *ACVR1B/ACVRL1*. The authors concluded that genetic variants could contribute to short-term UA metabolism. This is an important concept that has garnered considerable attention. The significance of this small pilot study is the acute challenge, as dietary behavior is wrought with intermittent sugar binges followed by abstinence and relapse.

Preclinical work over the past year has seen a report identify a significant sex disparity between male and female mice subjected to high-fat feeding (60% calories derived from fat) in the context of hyperuricemia and allied metabolic dysfunction [9]. This report clearly reveals the absence of hyperuricemia in female C57Blk/6J mice during and at the conclusion of 32 weeks of high-fat feeding. The female mice demonstrated significant weight gain versus time on diet as well as elevated markers of liver damage (ALT and AST) seen with NAFLD, impaired glucose tolerance, elevated fasting glucose, circulating insulin, free fatty acids, triglycerides, cholesterol, and cholesteryl esters. However, unlike male C57Blk/6J mice on the same diet [1], the obese female mice did not demonstrate elevated plasma UA when compared to age-matched lean controls. This finding reveals that a one-size fits all approach to obesity and UA is not appropriate and serves to identify a sex disparity that needs to be recognized, especially in future preclinical work designed to address the question of causation versus correlation for UA and metabolic dysfunction.

A considerable consequence of obesity/metabolic dysfunction is the accumulation of fat in the liver leading to nonalcoholic fatty liver disease (NAFLD) and nonalcoholic steatotic hepatitis (NASH). A published report (in abstract form) using a humanized mouse model whereby loss of function mutation was localized to the ABCG2 UA transporter found it to be heavily expressed in the intestine [20]. The mutant male mice were hyperuricemic due to loss of capacity to export UA from the intestinal epithelium. Interestingly, these hyperuricemic mice also displayed elevated plasma glucose and insulin levels as well as fat deposition in the liver (NAFLD) similar to that seen in models of high fructose administration. These results serve to reiterate the complexity of the relationship between circulating UA levels and metabolic dysfunction as hyperuricemia was induced in these mice by altering clearance of UA and not by elevating production via diet-induced increases in the XOR substrate xanthine. This report again affirms the crucial need for further exploration of this phenomenon.

### 4. UA and Hemolysis

As previously mentioned, the sole enzymatic source of UA is XOR and thus alteration in XOR expression/activity is one point of determining the concentration of UA in the circulation. Previous studies have shown that iron ($Fe^{2+}$ and $Fe^{3+}$) either free in solution or complexed to a silica scaffold or EDTA can induce expression of XOR [4]. This observation when coupled with observations of an association between ferritin and hyperuricemia [3] begs the question: How does iron overload affect circulating UA levels and why? One of the most common factors in iron overload is hemolysis. Hemolysis is seen in an array of pathologies (e.g., sickle cell disease, thalassemia, venom exposure, sepsis, etc.) and can be induced iatrogenically (e.g., cardiac bypass surgery, trans-aortic valve replacement (TAVR), extracorporeal membrane oxygenation (ECMO), etc.). However, little if any data are available for assessing the impact of this hemolysis on UA levels or, for that matter, XOR levels. In the past year, a report (published abstract, with manuscript in review) demonstrated that a murine model of intravascular heme crisis resulted in rapid and profound elevation (20-fold) in circulating XOR as well as a substantive increase in circulating UA [21]. The study further showed that systemic pharmacologic inhibition of XOR with febuxostat or liver-specific XOR knockout diminished survival suggesting: (1) the source of the elevated

XOR in the circulation was the liver and (2) XOR was protective. Biochemical analysis of the interaction between XOR and free hemoglobin revealed an XOR-dependent catabolism of the hemoglobin, binding and further catabolism of the released hemin, and a UA-dependent chelation of the subsequently released iron. The authors also revealed a hemin- and/or free-iron-mediated induction of XOR in hepatocytes followed by an exocytosis process resulting in the release of XOR into the culture medium. When taken together, this cascade of events revealed that hemolysis resulted in upregulation of hepatocellular XOR, released into the circulation where it catabolized the hemoglobin, and free hemin while the UA that was generated could chelate the released iron and hold it in a less redox-active state and thus diminish the toxicity to the vasculature [22] (Figure 2). While this study was focused mostly on XOR, it did reveal an important antioxidant role for UA in the context of hemolysis as well as validating previous work indicating that iron may play a key role in the production of UA.

$$Fe^{3+} + O_2^{\bullet-} \rightarrow Fe^{2+} + O_2$$

$$Fe^{2+} + H_2O_2 \rightarrow Fe^{3+} + OH^- + {\bullet}OH$$

**Figure 2.** UA chelates iron and prevents redox reactions. With a stoichiometry of 2:1, UA can chelate free iron and hold it in a manner (**upper panel**) that diminishes its capacity for redox reactions (**lower panel**). Examples of these redox reactions include the Haber–Weiss and Fenton reactions catalyzed by ferric (uppermost reaction) and ferrous (lowermost reaction).

## 5. UA and COVID-19

During the course of the COVID-19 pandemic, identification of durable biomarkers for predicting outcomes has been pursued with urgency. During this process, UA was identified as a potential candidate. In the past year, significant progress has been made in the context of assigning associative characteristics between circulating UA levels and severity of outcomes. For example, a preclinical study using the Syrian hamster model demonstrated a greater-than-4-fold elevation in plasma UA levels from 3 to 20 days post viral injection [23]. Unfortunately, data were not collected from 0 to 3 days and thus the temporal relationship between onset of infection and plasma UA levels is not known. However, it does indicate a positive correlation between infection and UA levels. This same trend was seen clinically as a study from a cohort of 1523 patients from a COVID-19 task force in Japan revealed a significant association between elevated circulating UA levels and/or a history of hyperuricemia and the risk for invasive mechanical ventilation (IMV) [24]. Countervailing this trend, a clinical study in which UA levels were assessed upon admission revealed diminished UA levels in patients hospitalized for COVID-19 compared to healthy controls [25]. This decrease in UA was significant regardless of the severity of disease (ICU vs. non-ICU) referrals. However, this study also suffers from limitations as single samples were analyzed from blood taken upon admittance. The authors do speculate that one potential explanation for their results countervailing reports of elevated UA in the context of COVID-19 may be the timing of the sample acquisition. Specifically, they suggest that the severity of the disease requiring hospitalization and furthermore

immediate referral to the ICU may coincide with COVID-19-associated hemolysis and thus diminish UA concentration due to UA–iron or UA–heme interactions and/or production of UA-related oxidation products. Interestingly, another report during 2022 also observed a considerable degree of hypouricemia upon admittance as well as significant correlation between a poor prognosis and decreased circulating UA in ICU patients [26].

In conclusion, while there were very few basic biochemistry/biology publications focused on UA in 2022, there were several reports that leaned heavily on basic concepts related to the biologic activity of UA in the context of metabolic and hemolytic disease as well as COVID-19. These reports serve to move the field forward by utilizing new genetic models for studying UA, discovering new protective antioxidant-associated reactions, and continuing to address the role of UA in the metabolic abnormalities linked to obesity and diabetes. As such, we look forward to 2023 and the new information that it will bring!

**Author Contributions:** Conceptualization, E.E.K. and R.D.K.; writing—original draft preparation, E.E.K. and R.D.K.; writing—review and editing, E.E.K. and R.D.K. All authors have read and agreed to the published version of the manuscript.

**Funding:** NIH R01 DK124510-01 and R01 HL153532-01A1 (E.E.K.)

**Institutional Review Board Statement:** Not applicable.

**Informed Consent Statement:** Not applicable.

**Data Availability Statement:** Not applicable.

**Acknowledgments:** We thank the NIH for their support.

**Conflicts of Interest:** The authors declare no conflict of interest.

## Abbreviations

$H_2O_2$ (hydrogen peroxide), $O_2^{\bullet-}$ (superoxide), UA (uric acid), UOX (urate oxidase), XDH (xanthine dehydrogenase), XO (xanthine oxidase), XOR (xanthine oxidoreductase).

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
