# Peer review of "Urate Biology and Biochemistry: A Year in Review 2022"

_2813-4583, doi:10.3390/gucdd1030011_

Round 1
Reviewer 1 Report
In the manuscript, “Urate Biology and Biochemistry: A Year in Review 2022,” authors provide a thorough summary of advances in urate biology and biochemistry reported in 2022. Authors present a strong argument that even with the addition of this novel information, much is still to be discovered in the field of urate biology. The manuscript in its current form is well structured but could benefit from some of the criticisms listed below in order of importance.
1. The summaries given in some sections are much more detailed than in others. Thus, the report on alterations in the microbiome of Uox-KO mice (page 3, line 113) need to be expanded upon to discuss some of those alterations (shifts in phyla, introduction of different microbial communities, etc). Additionally, in the following section, more details regarding the SNPs (page 3, line 127) could also enhance the manuscript. For example, it may be beneficial to list either the SNPs themselves or the adjacent genes, or both.
2. When discussing the preclinical work involving the mice fed high-fat diet, it is important to clarify the differences observed between male and female mice as “sex differences,” and NOT “gender differences.” These terms are not interchangeable. Differences being reported are a result of the biological sex and thus should be named appropriately (page 4, lines 133 and 144.)
3. Authors list two different “normal” ranges for serum urate in humas (page 1, line 33 and page 3, line 86.) These ranges are not consistent. Please provide justification for using these two different ranges, as well as citations for this information. It may also improve the manuscript to list the serum urate ranges for reported men and women separately, to enhance the argument regarding sex differences on page 4.
4. On page 2, authors discuss the abundance of variables published for studying obesity and metabolic syndrome in mice. Please specify which model is used for the male and female mice described in lines 63-65.
5. Finally, there are a few line editing issues that need to be addressed:
a. Page 3, lines 90-91 – a verb is missing in this sentence
b. Page 3, line 111 – hyperuricemia should be hyperuricemic
c. Page 5, line 199 – a verb is missing in this sentence
d. Page 6, line 291 – the year listed is 2024, please correct
e. Page 6, line 227 – please correct to “address the role of UA”
Author Response
We thank the reviewers for their critical evaluation of our review and for their constructive suggestions to strengthen our manuscript. Please find below a point-by-point response to their concerns:
Reviewer 1: In the manuscript, “Urate Biology and Biochemistry: A Year in Review 2022,” authors provide a thorough summary of advances in urate biology and biochemistry reported in 2022. Authors present a strong argument that even with the addition of this novel information, much is still to be discovered in the field of urate biology. The manuscript in its current form is well structured but could benefit from some of the criticisms listed below in order of importance.
Response: We thank the reviewer for their kind words.
- The summaries given in some sections are much more detailed than in others. Thus, the report on alterations in the microbiome of Uox-KO mice (page 3, line 113) need to be expanded upon to discuss some of those alterations (shifts in phyla, introduction of different microbial communities, etc). Additionally, in the following section, more details regarding the SNPs (page 3, line 127) could also enhance the manuscript. For example, it may be beneficial to list either the SNPs themselves or the adjacent genes, or both.
Response: We now include the 2 major phyla that were altered in the Uox KO mice. they did identify 2 phyla that were altered and we now include that information. In addition, more detail is now added in the following section as referred to above.
- When discussing the preclinical work involving the mice fed high-fat diet, it is important to clarify the differences observed between male and female mice as “sex differences,” and NOT “gender differences.” These terms are not interchangeable. Differences being reported are a result of the biological sex and thus should be named appropriately (page 4, lines 133 and 144.)
Response: Indeed, you are correct and these have been changed to sex.
- Authors list two different “normal” ranges for serum urate in humas (page 1, line 33 and page 3, line 86.) These ranges are not consistent. Please provide justification for using these two different ranges, as well as citations for this information. It may also improve the manuscript to list the serum urate ranges for reported men and women separately, to enhance the argument regarding sex differences on page 4.
Response: Yes, we were referring to the upper portion of the male and female ranges when discussing urate-mediated XOR inhibition. We have altered the narrative to eliminate the second range and simply state that urate can inhibit XOR at physiological levels.
- On page 2, authors discuss the abundance of variables published for studying obesity and metabolic syndrome in mice. Please specify which model is used for the male and female mice described in lines 63-65.
Response: Thank you for catching this. We have now included both the diet and mouse strain in the narrative.
- Finally, there are a few line editing issues that need to be addressed:
- Page 3, lines 90-91 – a verb is missing in this sentence
- Page 3, line 111 – hyperuricemia should be hyperuricemic
- Page 5, line 199 – a verb is missing in this sentence
- Page 6, line 291 – the year listed is 2024, please correct
- Page 6, line 227 – please correct to “address the role of UA”
Response: Thank you, these corrections have been made to the new document.
Reviewer 2 Report
I read this review with great interest and find it a good and easy way to get updated on a quite complicated topic. However, I have a few minor comments.
1
In Introduction, line 67-69 it is stated: "For example, the models used thus far to interrogate this issue have consisted of genetic (ob/ob, db/db and pound mouse) obe-68 sity and diet-induced obesity using various diets from the 42% to 60% calories derived 69 from fat to high-fructose or fructose in the drinking water"
I would like to see references to this statement
2
In the section UA and hemolysis different causes to hemolysis are presented (line 167-171). Here I would recommend a I would recommend a more overall, structured description of the casuses to hemolysis - I think there are important causes missing while at the same time the description of iatrogenic hemolysis is quite detailed.
3
The section on UA and hemolysis ends with figure 2. I have some problem to fully understand it. I would recommend a bit more explanation in the figure legend but also a referral to the figure in the text.
4
Typos
I miss an abbreviation for uric acid in the text, I can only find one in the abstract
Line 111-112: "As expected, these mice 111 are hyperuricemia and suffer from urate nephropathy." Should it read "hyperuricemic"?
Line 219: "Interestingly, another report during 2024 also observed a considerable degree...". Should it read 2022?
Author Response
We thank the reviewer for their time and efforts!!
Reviewer 2: I read this review with great interest and find it a good and easy way to get updated on a quite complicated topic. However, I have a few minor comments.
Response: We thank the reviewer for their kind words.
- In Introduction, line 67-69 it is stated: "For example, the models used thus far to interrogate this issue have consisted of genetic (ob/ob, db/db and pound mouse) obe-68 sity and diet-induced obesity using various diets from the 42% to 60% calories derived 69 from fat to high-fructose or fructose in the drinking water" I would like to see references to this statement.
Response: We have slightly reworded this sentence in the Introduction as well as added the requested references.
- In the section UA and hemolysis different causes to hemolysis are presented (line 167-171). Here I would recommend a I would recommend a more overall, structured description of the casuses to hemolysis - I think there are important causes missing while at the same time the description of iatrogenic hemolysis is quite detailed.
Response: We agree that there is an imbalance in the narrative and have altered our wording to deliver more equity in a manner that does not focus on hemolysis but on the impact of heme on urate.
- The section on UA and hemolysis ends with figure 2. I have some problem to fully understand it. I would recommend a bit more explanation in the figure legend but also a referral to the figure in the text.
Response: We have rewritten this legend to more clearly describe the figure.
Typos
I miss an abbreviation for uric acid in the text, I can only find one in the abstract
Line 111-112: "As expected, these mice 111 are hyperuricemia and suffer from urate nephropathy." Should it read "hyperuricemic"?
Line 219: "Interestingly, another report during 2024 also observed a considerable degree...". Should it read 2022?
Response: We thank the reviewer for catching these. All typos have been addressed.